# Reasons and External Factors That Influence Access to University and Job Placement Programs for Individuals with Intellectual Disability

**DOI:** 10.3390/bs13090745

**Published:** 2023-09-06

**Authors:** Ingrid Sala-Bars, Anabel Moriña, Ana Casas, Lucía Van Der Mel

**Affiliations:** 1Facultad de Psicología, Ciencias de la Educación y del Deporte Blanquerna, Universidad Ramon Llull, 08022 Barcelona, Spain; 2Departamento de Didáctica y Organización Educativa, Universidad de Sevilla, 41004 Sevilla, Spain; anabelm@us.es (A.M.); lvandermel@us.es (L.V.D.M.); 3Facultad de Educación y Deporte, Universidad de Deusto, 48007 Bilbao, Spain; ana.casas@deusto.es

**Keywords:** IDD, higher education, transition, inclusive education, workplace inclusion

## Abstract

Despite the rapid growth in inclusive university programs, access to inclusive higher education is still limited for students with intellectual disability (ID). This article explores the perspectives of 34 students with ID on their motives for accessing the inclusion and job placement programs at three Spanish universities and the external factors that contributed to their studying at the university. The study used a qualitative methodology based on a phenomenological approach using semi-structured interviews that had previously been validated and piloted. The data were analyzed using an inductive category and code system. The results addressed four questions: What is the participant’s academic pathway? What is their job profile? What are their reasons for studying at the university? What are the external factors that influenced their studying at the university? The study concludes that higher education can be an invaluable tool to foster the workplace inclusion of individuals with ID and promote their independent living. Furthermore, the family, organizations, and third-sector entities, as well as collaboration among them, emerged as key contextual factors for access to higher education and the personal and professional development of individuals with ID.

## 1. Introduction

Inclusive education is a basic human right and the foundation for a more just and equitable society, as recognized by various international organizations such as the European Agency for Special Needs and Inclusive Education, the United Nations, and UNESCO. While the right to inclusive education during compulsory schooling has been recognized for many years [1,2], access to higher education has historically been difficult for people with disabilities [3]. However, many countries have recently reaffirmed their commitment to the goal of more inclusive education by adopting the so-called Incheon Declaration [4] and its Framework for Action to achieve Sustainable Development Goal 4: “Ensure inclusive and equitable quality education and promote lifelong learning opportunities for all”. This declaration, promoted by UNESCO, advocates urgent action to transform and improve the quality of life of individuals through a new vision of education. This visionary approach emphasizes the critical importance of ensuring inclusive and equitable education at all levels, including not only compulsory education but also higher education. It highlights the need to remove barriers that limit access to education and emphasizes the promotion of equal opportunities for all individuals, regardless of their socio-economic background, gender, ethnicity, or disability. In addition, this new educational paradigm places particular emphasis on the development of relevant 21st-century skills, such as critical thinking, creativity, problem-solving, and collaboration, to enable students to effectively face the challenges of the modern world [5].

In recent years, the diversity of the student population on university campuses has increased, reflecting a commitment to inclusive education [6]. In Spain, approximately 1.5% of university students have some form of disability [7]. Despite this progress, the construction of an inclusive university remains a challenge, especially for students with intellectual disability (ID) [1,8] (a term according to the current model of the American Association on Intellectual and Developmental Disabilities [9]).

Until recently, access to higher education for students with ID has been limited [1]. However, despite their relatively recent access and participation in the university context, there is already a significant amount of international literature highlighting the important benefits of this experience in improving the quality of life of people with ID [2,3,10] (among others). In the Spanish context, there are no official data on the experiences and access of students with ID to higher education, although the number of university programs aimed at this population is increasing every year [11]. Given this lack of data, it is imperative to study and understand the transition process of these students to the university, with the aim of improving their access and experience in the university environment, thus allowing them to benefit from the positive impact that the university can have on their quality of life.

This study aims to explore the access of students with ID to higher education through a qualitative methodology that captures the experiences and perspectives of the students themselves. The aim is thus to identify key factors that influence their access to higher education and to gather relevant information to improve and facilitate their transition processes to university. Consequently, the study aims to improve our understanding of the experiences and access to education for Spanish students with intellectual disabilities (ID), recognizing the existing data gap in this area.

All in all, the research question underpinning the study is presented, delving into the existing knowledge about the inclusion of students with ID in university, as well as the transition processes to this academic stage. The methodological design and results of the study are described. The article concludes with a discussion and conclusions, as well as limitations and future research.

### 1.1. Inclusion of Students with ID at the University

Access to higher education for people with ID is a reality that has become more common in recent years in Australia, Europe, and North America [1,12,13]. This growth is providing new contexts for learning and interaction [1,14] that aim to prepare people with ID for greater inclusion in community life [13]. Spain is no exception. The presence of students with disabilities in university classrooms has been steadily increasing since the Organic Law on Universities of 2001 and the General Law on the Rights of Persons with Disabilities and their Social Inclusion of 2013, which guarantee equal opportunities to university students with disabilities [15,16]. In the specific case of students with ID, since 2004, when the Autonomous University of Madrid and the Prodis Foundation spearheaded the first pilot experience offered to eight students with ID at the university [17], the number of universities offering courses aimed at this student population has increased every academic year. With the recent approval of Organic Law 2/2023 on the University System on 22 March 2023, it is expected that these programs will continue to grow since, for the first time, Spanish universities will have to promote access to university studies for people with ID by promoting programs adapted to their capacities. These programs for young people with ID are continuing education programs based on customized degrees, which are neither official nor nationally regulated, with the main objective of promoting employment training for young people with ID in ordinary work contexts [8]. Unlike in other countries such as the United States or Canada [18], these programs are not planned within a fully inclusive framework. Initially, most of them were designed within the so-called “substantially separate model”, in which students with ID attend classes only with other peers with disabilities. In this model, the curriculum focuses primarily on life skills and vocational training; however, students with ID may have the opportunity to participate in general social activities on campus and may be offered work experience in mainstream settings [19]. In recent years in Spain, these programs have tended to be situated within a “mixed/hybrid model”, where students with ID attend university classes with students without disabilities [19]. The majority of these programs have been launched within the framework of the Operational Program for Youth Employment 2014–2020, co-financed by the European Social Fund, and managed by the Spanish National Organization for the Blind (ONCE) Foundation.

As the number of these higher education programs for students with ID has grown, studies have shown that studying at university is beneficial for students with ID [10,20,21,22,23]. These benefits include increased employment opportunities, social inclusion, independent living, and improved quality of life. Specifically, one study showed that the participation of students with ID in inclusive classes, campus events, volunteer work, and prior work experience increased their likelihood of securing paid employment while in college and their success in the workplace after graduation [18,24,25,26]. There is also evidence that participation of students with ID in university programs has a positive impact on their independent living and social satisfaction after graduation. In this sense, a significant percentage of students with ID live independently after completing their program, and the majority of them are satisfied with their social life [10,18,20,22]. The university experience also seems to give them better social skills and greater self-determination [27,28]. Furthermore, Alqazlan [19] concluded that participants in inclusive mixed higher education training models in which they share classes with students without ID were more likely to report more benefits, especially gains in social skills, compared to those in the separate model.

The participation of people with ID in higher education is not only beneficial for them but also for society in general [1]. This participation can reduce their dependence on government services, change others’ perceptions of their abilities, and increase awareness of their potential contributions to the community [1,25]. To ensure these benefits, it is essential that students with ID have access to university and a fully inclusive academic and social experience [1]. However, although more and more universities are taking steps to make the university experience more accessible and inclusive for students with disabilities, students with ID are still segregated in many university programs, and the staff of these programs are tasked with academic advising, teaching, and attention to accommodations, rather than using the entire university system to address their needs [2,19]. Some authors [1,3,29,30,31] warn that applying the medical model of disability to the inclusion of students with ID in universities could contribute to hindering their inclusion on campus and minimizing their impact on society.

### 1.2. Students with ID and the Transition to Higher Education

In recent years, legislative changes in favor of students with ID at university and the reported benefits of this experience have led to an increase in the number of these students, as their families and they themselves see university as a viable opportunity for them [22,32]. Students with ID have several reasons for enrolling in higher education programs, including perceived social benefits, a desire to improve their chances of finding and securing paid employment, and the opportunity to make new friends, especially friends without disabilities [19,33,34,35]. Plotner and May [36] compared the motivations for attending university among students with ID, students with moderate learning disabilities, and students without disabilities and found that the reasons for attending university were the same for all three groups: to learn new things, to earn more money, and to find a job. However, despite similar motivations for accessing higher education among different groups of students, a difference was observed in the perspective of students with ID compared to others. Unlike the other groups, students with ID did not see higher education as a way to pursue a specific career path. Instead, they valued the social benefits of the experience and the opportunity to gain greater independence in their lives. On the other hand, Causton-Theoharis et al. [37] identified the reasons why families supported their children’s participation in university programs. Specifically, families enrolled their children in higher education programs because they did not want them to stay at home after completing compulsory education and because they believed that these programs were important for the education of adults with ID.

However, the lack of cooperation between the staff of higher education programs, secondary schools, teachers in the programs, community agencies, and families was identified as an obstacle to the transition and engagement in higher education programs for students with ID [38].

A lack of coordination between higher education programs for students with ID staff and secondary school teachers has been identified, which hinders students with ID’s understanding of the goals of university programs and the skills needed to succeed [39,40,41]. In order to improve access and inclusion of students with ID in universities, collaborative partnerships between universities, secondary schools, and vocational programs need to be established. Such collaboration would ensure that students with ID receive the necessary information and training prior to enrolling in universities so that they feel confident and equipped with strategies for success upon completion of secondary education [38,42].

Regarding families, they are usually unaware of the availability of these post-secondary education programs for their children due to a lack of information from both secondary schools and community agencies [43,44]. In this context, the active participation of students and their families in transition planning can play a fundamental role in the success of their experiences in higher education programs [38].

In Spain, this transition is not recognized as a stage that requires specific social and educational support, which makes it difficult to support young people with ID during this period. It has been reported that these adolescents have great difficulties in receiving support during the transition, which takes place after compulsory education in different schools [45].

Although there has been an increase in the number of people with ID at university, they still face significant barriers to accessing higher education compared to other students with disabilities [17,46]. As higher education programs for people with ID have grown, studies have examined the characteristics of the educational programs; the perceptions, attitudes, and training of faculty; the motivation and empowerment of students with ID; and the impact of these experiences on work [18]. However, there are few studies that use the voices of students with ID to explore their reasons for studying at university and the external factors that could contribute to removing barriers to higher education. Therefore, the purpose of this study is to use the narratives of students with ID to identify the contextual factors that facilitate their access to university, taking into account a systemic perspective.

## 2. Materials and Methods

This study is associated with three university job inclusion and placement training programs for youths with ID offered at three Spanish universities. The study used a qualitative methodology based on the phenomenological approach [47] to answer four research questions:What is the participant’s academic pathway?What is their job profile?What are their reasons for studying at the university?What are the external factors that influenced their studying at the university?

The phenomenological approach was used in the study because, as explained by Van Manen [47], it aims to study and understand these four research questions from the perspective of the student participants in the educational process. Within this approach, subjective meanings and experiences of individuals in relation to the proposed topic are crucial.

### 2.1. Description of the Programs

#### 2.1.1. Program at University 1

The objective of this program is to offer comprehensive job training with support in ordinary work settings and social and university inclusion for 15 youths with ID ranging in age between 18 and 30. Students enrolled in this program usually want to continue their academic and vocational training in order to find a job that can foster their transition to adulthood, increase their quality of life, and improve their employability, generating opportunities that promote a full, independent, and satisfactory life. This program is partially funded by the main families of students with ID. It is held across two academic years, each worth 25 ECTS (ECTS correspond to European credits; 1 credit is equivalent to 25 h, with 10 class hours and 15 h of independent student work), with a total of 13 subjects (Current Events, Culture and Society I and II; Personal Development and Autonomy I and II; Accounting and Finances I and II; Computer Science I and II; Social and Communicative Skills; Introduction to Business; Practicum).

#### 2.1.2. Program at University 2

This program, financed by the ONCE Foundation and the European Social Fund (ESF), exclusively targets youths with ID between the ages of 18 and 29 who want to improve their social and job skills through internships in ordinary enterprises. The program, which lasts one academic year (32 ECTS), has the capacity for 17 students and is held in a university setting to foster coexistence with other students. It consists of eight modules (Healthy Lifestyle; Information and Communication Technology; Communicative Skills; Basic Economics and Social Mathematics; Basic Job-Related Concepts; Tutoring; Practicum; and Final Project), and its ultimate goal is to improve the autonomy and academic training of youths with ID, as well as to prepare them for job placement in ordinary settings with support. The course also seeks to offer them educational experiences integrated into the university community and to provide comprehensive, personalized training that enables them to fully participate in community life.

#### 2.1.3. Program at University 3

This program, also financed by the ONCE Foundation and the ESF, has the same purpose as the program at University 2. The course targeted at youths with ID consists of 30 ECTS taken in one academic year. It also has the capacity for 17 students and is held in the Faculty of Education. The training program is organized into two main thematic sections: general training for employability and social inclusion, with five modules (social and emotional skills and cognitive training, personal wellbeing for social and job inclusion, digital competencies, tutoring, and university life) and specialized training in six modules (specific training in social and work orientation, mathematics and administrative tasks, work activities, tutoring, internships in companies and university life). Throughout the subjects, transversal inclusive classes are held with students in the different degree programs of the Faculty.

### 2.2. Participants and Recruitment

To recruit the participants, the three first authors of this article who are responsible for those training programs contacted students who were currently studying in the three training programs. An in-person meeting was held at each of the participating universities to present the study, its goals, the method used, and the time required for the interview. They were reminded that participation was free and that whoever was interested in collaborating on the project should think about it and communicate their final decision. Of the 43 students, 34 ultimately agreed to participate in the study: 13 studied at University 1 (P1–P13), 10 at University 2 (P14–P23) and 11 at University 3 (P24–P34).

The general information on the participants is contained in Table 1. Twenty-two participants (64.7%) were women, and 12 were men (35.3%). Their mean age was 23.61 years, and the standard deviation was 2.808, while the age range with the highest percentage (44.1%) was 20 to 22 years old. The majority of them have been diagnosed with a mild intellectual disability (61.8%), and the majority of families were made up of the mother, father, and sibling(s) (58.9%).

### 2.3. Data Collection

A script was developed for a semi-structured interview that included five dimensions: general information, academic information, work experience, motivations, and contextual factors. Some examples of questions included in the script: Who are you currently living with? Why are you taking this course? Do you think your family influenced your decision to go to college? If so, why? Do you think your friends influenced you to go to college? If so, why?

Before the interviews were conducted, the script was validated by Easy Read experts to ensure that the questions were cognitively accessible. The interviews were also piloted with two students from previous editions of the training program at Universities 1 and 3. After the pilot, minimal changes were made to the wording of some items to make them easier to understand.

The interviews were conducted in person by the first three authors of the article at the faculties where the courses were taught. The average duration was 20 min. Each interview was recorded, transcribed, and returned to the participants so that they could change anything they felt needed to be corrected. The interviews were conducted in the Spanish language. Subsequently, for this paper, the authors translated quotes into English, and a native translator reviewed the entire paper.

### 2.4. Data Analysis

An ad hoc analysis system was used to analyze the findings, following the recommendations of Miles and Huberman [48]. After all interviews were transcribed, a thorough reading of the data was conducted, and a system of categories and codes was developed (Table 2). Initially, one of the authors of the article created the system, which was then reviewed and expanded by the other co-authors. This system facilitated the interpretation of the data and allowed for the coding of all interviews.

### 2.5. Researchers Positionality

The four authors of this manuscript are women, one of whom has a hearing impairment. Our research focus has always been inclusive education and disability at different levels of education. The first three authors have been directors of vocational training programs for young people with disabilities at their respective universities for many years. The last author is a researcher and support staff member of the University 3 program.

We are committed to improving the academic and professional lives of people with disabilities and to using qualitative research methods to give voice to marginalized groups, such as people with disabilities.

### 2.6. Validity and Reliability

A qualitative methodology was chosen to give voice to students with ID from the three programs. We were interested in knowing and understanding in depth their explanations of the research questions that guided the study. In order to guarantee validity, the methodological design took into account aspects such as (a) the sample (since the students of the program were the participants of the study, this guaranteed that the phenomenon of the study could be analyzed from their perspective), (b) the creation of the interview script by all the authors of the article, (c) the validation (by experts of the intellectual disability field and to adapt materials to an easily readable format) and piloting of the interviews (with former graduates of the same program in previous editions), (d) the triangulation of the system of categories and codes by all the authors, and (e) the rigorousness of the analysis techniques and the systematic process of transcription. In addition, to ensure reliability, the research process was intended to be coherent, transparent, and clear, with the four authors working collaboratively and reflectively at all times. Throughout the process, we also used reflexivity to critically examine our own biases as we analyzed the data.

### 2.7. Ethical Aspects of the Research

Initially, the study was approved by the ethics committee of the university affiliated with the first author. This approval complied with the provisions of Organic Law 3/2018 on the Protection of Personal Data and Digital Rights, as well as the parameters established by the General Data Protection Regulation (EU) 2016/679, in conjunction with the relevant parallel legislation. Subsequently, an informed consent form, validated by professionals specialized in Easy Read accessibility, was used to inform participants of the research objectives. This document explained the voluntary nature of participation and emphasized the unwavering commitment to maintaining the confidentiality and privacy of their data throughout the study. In addition, participants received transcripts of the interviews for review. In lieu of actual names, alphanumeric codes (e.g., P1, P2, and P3) were used to ensure anonymity.

## 3. Results

### 3.1. What Is the Academic Path of Young People with ID?

During their pre-university education, the most common academic path of the students with ID was to have attended ordinary schools with specific support. In primary school, the majority of participants (n = 32) attended regular schools, while one student had a split education (combining attendance at a regular school and a special school during the academic year) (P18), and another attended a special school (P14). Of the students who attended regular schools, 15 reported having received special educational assistance during their schooling.

At the secondary level, the majority of participants (n = 25) studied in regular schools, while three participants reported that they spent the first two years of secondary school in regular schools and the last two years in special schools. Six participants completed their entire secondary education in special schools. Of the 25 students who completed all of their secondary education in regular schools, 17 reported receiving special educational assistance during this time. Regarding whether they had completed compulsory secondary education, 21 participants said that they had not, while 13 said that they had.

At the end of secondary school, the majority of participants (n = 27) indicated that they had received basic vocational training (which does not require completion of compulsory secondary school). Of these 27 participants, 14 reported that they had received this training in special schools. Two participants mentioned that they had completed intermediate training, and one participant said that he had studied up to the second year of the baccalaureate. Finally, seven participants indicated that they had not completed any type of vocational training.

### 3.2. What Is the Job Profile of Young People with ID?

Participants in the study reported different types of work experience as well as the reasons why they thought they could or could not obtain a job. The majority of participants had no previous work experience (n = 25). Of the nine youth who had some type of paid work experience, one had a paid job for a few days (P15), another worked in his parents’ business (P16), and two others had paid internships (e.g., P33).

On the other hand, participants were asked why they thought they had no work experience. One thought that she had a job opportunity because of her aptitude and skills for the job (“Because I work well and the association helped me find a job”, P13). The main reason participants reported not finding a job was that they had not looked for a job (n = 11). For example, participant 11 stated, “I haven’t started looking for work yet. Another of the most common reasons for not finding work (n = 10) was lack of education. Specifically, they said that they did not have a paid job because they needed to study more (“Because I need to study more” (P19)), they did not feel prepared (“I did not feel prepared” (P20)), or their current goal was to obtain an education in order to work later (“Now my goal is to work” (P26)). Another participant said that she preferred to work as a volunteer for now “because I like volunteering more” (P23). Lack of skills related to work (n = 1) or job search (n = 1) were two other reasons given by the participants for not having any paid work experience, as the following two participants pointed out: “Computer work is very complicated” (P17) and “I would like to be taught how to find a job” (P31).

### 3.3. What Are Their Reasons for Studying at the University?

The participants in this study attend the training programs for different reasons. The main reason cited by half of the participants (n = 17) is that they want to find a job to enter the labor market and earn a salary that would allow them to live independently. Of the three universities where programs were held, most of the participants who cited this argument were at University 1, as 11 of the 13 participants clearly know that they are studying at the university to obtain a job: “Because I want to find a job in the future so that I can live on my own, live an independent life” (P8).

Other participants’ (n = 10) purposes for attending these programs were to continue their education and to learn new things and new skills that would allow them to feel prepared for their lives and future jobs. This argument was shared mainly by the students at University 3: “I want to be academically educated. Also, I would be excited to go to a university” (P25).

Meeting other people was the reason given by four participants, namely one from University 2 and three from University 3. It was important for them to study at the university because it allowed them to build social networks, as they said they had no friends: “Studying, being educated, and meeting people because I don’t go out in my neighborhood” (P26).

Four other participants did not want to participate in the training programs out of their own motivation but because their families or third-sector professionals recommended it to them in order to improve their personal skills, such as autonomy: “Well, I’m doing this course because my mother told me that it’s very important to be more autonomous” (P24).

Three participants argued that their main motivation was that the program offered the possibility of doing professional internships: “Because this course offers internships” (P19). A final motivation, mentioned only by one participant from University 3, was the opportunity for personal growth that he could gain through this training: “I grow as a human being” (P28).

### 3.4. What Are the External Factors That Influenced Their Study at the University?

The family was the factor that all the participants identified as decisive for their access to education, with the same results in all three universities. While all the participants mentioned the family as a whole as a decisive factor, “My family encouraged me to do this course. They told me: come on, if you want to work, take this course and finish it, you will be very well prepared for the future”. (P7), some participants mentioned the support of specific members of the nuclear family, such as their parents (n = 14) or siblings (n = 5), while others referred to extended family members, such as aunts and uncles (n = 3), grandparents (n = 3), or cousins (n = 1).

When participants referred to their parents, mothers (n = 7) tended to play a more proactive role: “Because my mom basically showed us everything that she thought could help us in the future and could be good for us to get a job or something that would be good for us” (P6). For one participant (P25), this proactivity in seeking support and resources came from his sister: “She was the one who got involved and enrolled me in a lot of places, so many that even she forgot where I was enrolled. She made me do things. She took responsibility for finding information about disability, so he considers her my mother and father”.

Other participants (n = 7) stated that both their father and mother provided the necessary support, directly or indirectly: “…My family helps me…” (P13). Similarly, one participant highlighted his sister (P14) and another his cousins (P33) as the people who recommended him to participate in the program because they had done so in previous years: “Also, I could know the university where my sister is studying” (P14).

Thus, at all three universities, the family provided both support and encouragement and shared the thrill and excitement that participants felt when they began their university studies. “When I realized I got in, I started crying because I was so excited to get in. My family was really happy” (P33). Some students highlighted other ways in which their family influenced them to study. Sometimes the support was more focused on practical matters (n = 10), such as the paperwork needed for the admissions process: “They told me what to put in or what documents not to put in, but I wrote the application” (P34), as well as academic and career advice: “For example, my mom helps me a little bit with the computer, she helps me with everything I ask, she explains it to me 500 times so I understand it” (P31). This type of support was not often mentioned by students at University 1 (only one participant mentioned it), whereas it was important for students at Universities 2 and 3. In other cases, the support they received from their families was more emotional and moral, a source of confidence: “Well, my family supports me a lot. In all the decisions I’m considering, the good ones and the not so good ones, they’re always there and they never leave me alone” (P21).

In addition, three participants who have a partner highlighted how this person helped them to access university, especially emotionally, by encouraging them to take the course: “They helped me prepare for work and it was their idea to tell me to go to university” (P15).

Another external factor that influenced access to university was third-sector associations or institutions. In fact, the majority of participants belonged to an association, and they were the ones who directly informed the participants (Universities 1 and 3) or their families (University 2) about the existence of these training programs and the opportunities they offered. In some cases, the professionals from these organizations offered assistance with the application paperwork and even prepared the participants for the entrance exams (CV, interviews, or written tests): “I even rehearsed a little bit what the interview would be like” (P34).

To a lesser extent, teachers from compulsory or post-compulsory education also influenced participants’ access to university programs. With no differences between the three universities, 19 participants were academically supported by their teachers, who told them about these courses and encouraged them to take them and find a job: “All the teachers I’ve had have encouraged me a lot and told me that if you want to work, go ahead and take this course and get ahead in life” (P7).

Another factor in studying at university, although less prominent, was friends. In fact, more than half of the participants (n = 20) at all three universities believed that studying at the university was not conditioned by their friends because they did not share it with them: “They (my friends) didn’t know anything about what I was going to study here, I told them when I was already here” (P24). However, the other participants believed that their friends had supported and encouraged them to study at the university. One of the participants even studied because a friend told her about it: “My friend said go ahead, you can do it, of course you’ll get into the course” (P16).

Finally, other external professionals played an important role in the participants’ decision to study at the university, albeit to a lesser extent. Five participants (4 from University 2) identified their psychologists as key to their success in reaching the university, and one participant indicated that his coach had helped him: “He encourages me, he supports me, he tells me to keep going and that I can do it” (P11). Similarly, although the interview did not ask about personal factors, two students from Universities 1 and 3 believed that they themselves were actually the most important factor in studying at university: “Because I thought I might be able to break the barriers, because I thought I was a girl in a bubble, I got into a bubble that I never left and now I’ve fought and that’s allowed me to move forward” (P28).

## 4. Discussion and Conclusions

Through the narratives of 34 university students with ID, this study sought to identify the reasons and contextual factors that contributed to or facilitated their access to university. In terms of their academic pathways prior to university, the results suggested that the most common pathway for students with ID during compulsory education was to attend mainstream schools with special support. However, the majority of these students did not graduate from secondary school, indicating that there are still major challenges in the educational system to ensure the inclusion and academic success of students with ID [49]. It is important to highlight the fact that some students had to change their educational modality during secondary school, moving from regular schools to special schools. This change illustrates the difficulties that still exist in ensuring inclusive education in secondary schools [50,51]. Furthermore, the majority of the participants completed their basic vocational training in special education schools, which, in line with Pallisera et al. [45], suggests that the educational options for these students may be limited and that there may be a lack of inclusion in ordinary schools at these grade levels. Therefore, we note that in our context, a significant number of these students come from special education institutions when they enter university. Establishing robust communication mechanisms with these institutions to facilitate access and improve the inclusion of these students in higher education is urgently needed [38,45].

In terms of the employment profile of young people with ID before entering university, the majority of participants had no previous work experience. This can be attributed to a variety of reasons, including the lack of inclusive employment opportunities that allow them to access the labor market [10,45,52]. Some participants also identified lack of training as one of the main reasons for their lack of paid work experience. This confirms the need to develop post-compulsory training and skill-building programs specifically for people with ID that can help prepare these young people to enter the labor market [19,43,52] (among others). Therefore, in our context, work experience does not seem to be a determining factor for university access. On the contrary, these students believe that education and training are essential for obtaining employment.

Regarding the reasons for studying at university, it is encouraging to see that the majority of participants are motivated by the desire to find a job that will allow them to be more economically independent. Higher education can be an invaluable tool to promote the inclusion of people with ID in the workplace and to support their autonomy and financial independence [19,35,36]. However, it is also important to note that other participants are motivated by the opportunity to acquire new skills and competencies, as well as the opportunity to undertake internships. These motives reflect the importance of higher education as a means of personal and professional development beyond the goal of securing a paid job. It is also noteworthy that some participants are motivated by the desire to meet other people and build social networks. This factor highlights the importance of higher education as a tool for the social inclusion of people with ID [1,33,35]. The positive impact of access to university for persons with ID is evident. Therefore, it is necessary to develop policies that ensure access and participation of persons with ID in higher education. This implies that universities must be able to provide opportunities and resources necessary for their academic, professional, and social success while promoting employment inclusion, financial autonomy, skills and competence development, and the formation of social networks. These measures will contribute to improving the quality of life and opportunities for people with ID, enabling them to reach their full potential and become active members of society.

Regarding the external factors that influenced the participants to study at university, the results highlighted the importance of family support in accessing higher education. The family clearly plays a crucial role in promoting the educational inclusion of people with ID and supporting their personal and professional development [37,38,43,44,53]. Some participants also received recommendations from third-sector professionals to participate in training programs, suggesting that third-sector organizations and institutions may also play an important role in promoting inclusive higher education and the development of personal and professional skills of people with ID. The collaboration of all these contextual factors is essential to ensure that people with ID have access to university [38,42]. These findings highlight the key role that families play in providing emotional support, guidance, and decision-making related to the education of their children with ID. Their active involvement can make a significant difference to their children’s access and success in higher education. It is important for families to be aware of the benefits and opportunities that higher education can bring to people with ID. Often, families may have doubts, concerns, or a lack of information about the feasibility and value of higher education for their children with ID. Therefore, effective ways should be sought to inform and raise awareness among families about the benefits and opportunities that higher education can bring. It is important to note that higher education offers not only the acquisition of academic knowledge but also opportunities for the development of social, emotional, and vocational skills. Higher education can enhance the autonomy, independence, and personal growth of people with ID. It can open doors to enriching learning experiences, building social networks, participating in extracurricular activities, and preparing for the workforce.

In order to ensure that families are informed and can adequately support their children with ID, it is necessary to establish effective channels of communication between educational institutions, third-sector organizations, and the families themselves. Information sessions, workshops, and educational resources should be provided to address the opportunities, challenges, and support strategies for accessing higher education. In addition, it is important to have trained and specialized staff who can guide and advise families in the decision-making process related to higher education.

In addition to the crucial role of families, it is important to highlight the role of the third sector as an agent that can promote autonomy in the transition of persons with ID to university. The third sector, consisting of non-governmental organizations, foundations, and associations, plays a crucial role in promoting inclusion and supporting people with ID in various aspects of their lives. Through the provision of information, advice, support programs, and awareness raising, these organizations contribute to empowering people with ID and facilitate their transition and inclusion in higher education. Collaboration with the third sector is essential to ensure that people with ID have access to the opportunities and resources necessary to succeed in their educational journey and reach their full potential.

In conclusion, the current study highlights that students with ID who are effectively pursuing higher education are primarily those who have completed their compulsory education in mainstream academic settings with support and have expressed a desire to further their education in order to acquire knowledge that would enable them to access employment opportunities and achieve a level of independence comparable to their peers, and whose transition has been facilitated by support from both family and third-sector organizations. However, significant challenges remain in ensuring the right of persons with ID to inclusive, quality education at all levels of education, enabling them to make a successful transition to the labor market. After completing compulsory education, people with ID aspire to continue their education and lead independent lives like the rest of society. However, they face significant barriers in accessing both inclusive educational environments and regularly supported work environments, in violation of their fundamental rights. There is a clear need to develop post-compulsory training and skill-building programs specifically for people with ID, which may help prepare the youths to join the job market. In this sense, higher education can be a valuable tool to foster workplace inclusion and promote independent living. However, the current configuration of secondary school hinders these students’ access to the university. Therefore, university programs based on the “substantially separate model” or the “mixed/hybrid model” may be an opportunity to break this structural barrier in the short and middle term and pave the way for the construction of a university where all students, regardless of whether or not they have a disability, can fit within an inclusive framework. Furthermore, third-sector organizations and entities, as well as collaboration among them, are key contextual factors for access to higher education and personal and professional development among individuals with ID. These findings may be useful in developing policies and strategies that promote inclusive higher education where students with ID can be successful.

### Limitations and Further Research

Although only 34 students with ID participated in this study, which may be a limitation, the voices of these participants are a good starting point for the educational system to reflect on how to improve the transition processes of this group to university. It is essential to include more people with ID in future studies, as well as other actors involved in the educational transition process (families, teachers, and professionals), to learn more about the influence of these contextual factors and to identify other key factors to eliminate the barriers that hinder their access to higher education. Future studies could also use other data collection techniques, such as photo elicitation or the photo–voice method. Finally, to verify the impact of these programs in meeting the expectations of students with ID, longitudinal studies could enhance knowledge about their academic, vocational, and social development over time.

## Figures and Tables

**Table 1 behavsci-13-00745-t001:** General information of the participants (N = 34).

**Gender**	**N**	**%**
Men	12	35.3
Women	22	64.7
**Age**	**N**	**%**
20 to 22	15	44.1
23 to 25	10	29.4
26 to 29	9	26.5
**Intellectual Disability Severity**	**N**	**%**
Mild	23	67.6
Moderate	6	17.7
No specified	5	14.7
**Other associated Disabilities**	**N**	**%**
ASD	2	5.8
Epilepsy	1	2.9
**Family members**	**N**	**%**
Mother and father	4	11.8
Mother, father and sibling(s)	20	58.9
Mother, father, sibling(s) and other members	2	5.8
Single parent with sibling(s)	4	11.8
Single parent without sibling(s)	3	8.8
Guardian	1	2.9

**Table 2 behavsci-13-00745-t002:** Category and code system.

1. General information	1.1. Age
	1.2. Gender
	1.3. Intellectual Disability Severity
	1.4. Family members
2. Academic pathway	2.1. Compulsory studies prior to the university
	2.2. Post-compulsory studies prior to the university
3. Work experience	3.1. Type of work experience
	3.2. Reasons for no work experience
4. Motivations for studying at the university	4.1. Job placement
	4.2. Further training
	4.3. Meeting people
	4.4. Keeping occupied
	4.5. Internships offered
	4.6. Personal growth
5. Contextual factors	5.1. Family
	5.2. Friends
	5.3. Teachers
	5.4. Disability association
	5.5. Outside professionals
	5.6. Partner

## Data Availability

The data presented in this study are available on request from the corresponding author.

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
