# Peer review of "Reasons and External Factors That Influence Access to University and Job Placement Programs for Individuals with Intellectual Disability"

_behavsci, 2023, doi:10.3390/bs13090745_

Round 1

Reviewer 1 Report (Previous Reviewer 1)

The authors did not make the proposed improvements în order the article to be proposed for publishing. Especially the improvement of the research methodology, thus, the article is rejected from publishing.

Moderate editing of English language required

Author Response

Response to Reviewer 1 Comments

(1) The authors did not make the proposed improvements în order the article to be proposed for publishing. Especially the improvement of the research methodology, thus, the article is rejected from publishing.

We appreciate your recommendation. As we mentioned in the first review, it is important to note that our study is based on a specifically designed qualitative methodology. The inclusion of a quantitative instrument would require a significant methodological change, necessitating a complete redesign of the entire study. It would also require the use of different types of instruments and methods for data collection and analysis. Given that the research has already been completed, it is not possible for us to make drastic methodological changes at this time. Maintaining methodological consistency is essential to ensure the validity and integrity of the results obtained so far.

In addition, it is important to emphasize that qualitative research has been recognized and valued for many years as an appropriate approach for exploring participants' subjectivity and achieving a deep understanding of the phenomenon under study. This approach allows us to capture the richness of nuances, perspectives, and individual experiences, providing a holistic and enriching view of the subject matter. Therefore, we believe that the use of qualitative methods is the most appropriate option to achieve the objectives of our study and provide a comprehensive and meaningful understanding of the collected data without compromising the integrity of the already established research design.

Reviewer 2 Report (Previous Reviewer 2)

Dear author(s)

Hope you are doing well. According to the review of this article, the corrections have been made.

Good luck

Author Response

Response to Reviewer 2 Comments

(1) Hope you are doing well. According to the review of this article, the corrections have been made.

Good luck

Thank you so much. Your suggestions have certainly helped us improve the paper.  

Reviewer 3 Report (New Reviewer)

Thank you for the opportunity to review this manuscript “Reasons and external factors that influence access to university and job placement programs for individuals with intellectual disability” This manuscript used a qualitative research method to explore the perspectives of students with intellectual disability on their motives for accessing the inclusion and job placement programs. 

Introduction: 

In the introduction, I would suggest the authors define or explain “Incheon Declaration” and its framework. More explanation would be helpful for readers to understand the context. 

The authors mentioned “The term ‘lifelong’ speaks directly to universities as institutions 37 and has made the challenge of building an inclusive university unavoidable” Could the authors explain why “lifelong” made the challenge of building an inclusive university unavoidable? 

The authors mentioned” Despite this progress, the construction of an inclusive university remains a challenge, especially for students the intellectual disability (ID).” I did not see why an inclusive university is a challenge, especially for students with ID. What does the existing literature say about it? 

I do not understand the meaning “despite their relatively recent inclusion.” Please rewrite.

The authors mentioned “By addressing this issue, the study aims to fill the existing gap in providing meaningful theoretical contributions.” What is the gap? What gap are you trying to fill in? Is “this lack of data” a gap?  

In the section “Students with ID and the transition to higher education” 

First, I do not understand “However, there were differences in the dimensions of these reasons, since students with ID were less likely to see higher education as the pathway to a specific career, while they were more likely to consider its social benefits like increased independence.” Please rewrite.

In the paragraph starting “However, the lack of cooperation among the…..,” I would suggest the authors utilize separate paragraphs to delve deeper into the concept of "barriers." The paragraph contains various barriers, as mentioned by the author, but the narrative lacks organization. 

Line 61, the authors mentioned “theoretical framework.” What theoretical framework does the study apply? If not, please remove theoretical framework. 

Methods: 

Please explain why the study used “phenomenological approach” in the text. 

I saw the authors said “ECTS” many times. What does ECTS stand for? Does it mean credits in college? 

Were the interviews conducted in English? If not, the authors should describe transcription and translation process including interpreters and translators. 

The authors should address researcher positionality. Researchers' ontological and epistemological beliefs play an important role in qualitative research.

The authors seemed to engage in multiple efforts for trustworthiness. There are a couple of questions. 

1.     The authors mentioned “the sample (since the students of the program were the participants of the study, this guaranteed that the phenomenon of the study could be analyzed from their perspective).” Researchers normally have biases or their own views to interpret qualitative data, did the authors use reflexivity to scrutinize their own biases when analyzing the qualitative data?

2.     “The creation of the interview script by all the authors of the article” As previously mentioned, it is essential for the authors to address researcher positionality, which includes the background and experiences of the researchers relevant to the study topic.

3.     “The validation and piloting of the interviews” How did the authors validate the interview scripts or the interviews? Piloting an interview script does not necessarily mean validation. Please exercise caution when using the term "validation."

4.     “The rigorousness of the analysis techniques and the systematic process of transcription.” 

After reading the data analysis section, I found it difficult to comprehend the specific steps used by the authors to analyze the data and establish the code system. Did the authors use the code system to code the data? Or? Please explain. 

5.     “Reading of all the results and analysis, and the detailed information about the methodological process followed to facilitate the transferability of the study” I do not understand what it means. 

Discussion and Conclusions:

The discussion is lengthy. The authors kept highlighting the point regarding the fact that some students had to change their educational modality during secondary school. However, this particular point seems to be unrelated to the main focus of the study (I believed the main focus of the study was centered around post-secondary options). To improve the clarity and relevance of the study, I would recommend that the authors provide a concise description that highlights the implications for secondary school, while shifting the primary focus towards university and job placement programs. I found the discussion on external factors and family support to be engaging and enjoyable to read.

Extensive editing of English language required

Author Response

Responses to Reviewer 3

Introduction

1- In the introduction, I would suggest the authors define or explain “Incheon Declaration” and its framework. More explanation would be helpful for readers to understand the context.

Thanks for your suggestion. A better explanation about “Incheon Declaration” has been introduced in the first paragraph. 

2- The authors mentioned “The term ‘lifelong’ speaks directly to universities as institutions 37 and has made the challenge of building an inclusive university unavoidable” Could the authors explain why “lifelong” made the challenge of building an inclusive university unavoidable? 

We have rephrased the explanation using different terms in hopes of improving clarity.

3- The authors mentioned” Despite this progress, the construction of an inclusive university remains a challenge, especially for students the intellectual disability (ID).” I did not see why an inclusive university is a challenge, especially for students with ID. What does the existing literature say about it?

Thank you for your suggestion. We have included relevant literature confirming this, and the following paragraph explains that access for people with intellectual disability is still limited. 

4- I do not understand the meaning “despite their relatively recent inclusion.” Please rewrite.

Thank you. We have rephrased the explanation using different terms in hopes of improving clarity.

5- The authors mentioned “By addressing this issue, the study aims to fill the existing gap in providing meaningful theoretical contributions.” What is the gap? What gap are you trying to fill in? Is “this lack of data” a gap?  

Thank you for your observation . We have rephrased the explanation using different terms in hopes of improving clarity.

6- In the section “Students with ID and the transition to higher education” 

First, I do not understand “However, there were differences in the dimensions of these reasons, since students with ID were less likely to see higher education as the pathway to a specific career, while they were more likely to consider its social benefits like increased independence.” Please rewrite.

Thank you for the comment. You are right, that sentence was difficult to understand. We have rephrased the statement to make it more understandable.

7- In the paragraph starting “However, the lack of cooperation among the…..,” I would suggest the authors utilize separate paragraphs to delve deeper into the concept of "barriers." The paragraph contains various barriers, as mentioned by the author, but the narrative lacks organization.

Thank you for your suggestion. We have rephrased the explanation using different terms in hopes of improving clarity.

8- Line 61, the authors mentioned “theoretical framework.” What theoretical framework does the study apply? If not, please remove theoretical framework. 

We removed it. Thanks.

Methods: 

9- Please explain why the study used “phenomenological approach” in the text. 

We have explained this in the text. 

10- I saw the authors said “ECTS” many times. What does ECTS stand for? Does it mean credits in college? 

Exactly, they are European credits at university. We will check this in the manuscript and clarify it.

11- Were the interviews conducted in English? If not, the authors should describe transcription and translation process including interpreters and translators. 

The interviews were conducted in Spanish. Subsequently, for this article, we translated them into English, and a native translator reviewed the entire paper. We have indicated that in the manuscript.

12- The authors should address researcher positionality. Researchers' ontological and epistemological beliefs play an important role in qualitative research.

We have added with our research positionality.

13- The authors seemed to engage in multiple efforts for trustworthiness. There are a couple of questions. 

Thank you for all the recommendations below. We have incorporated it into the manuscript.

  1.     The authors mentioned “the sample (since the students of the program were the participants of the study, this guaranteed that the phenomenon of the study could be analyzed from their perspective).” Researchers normally have biases or their own views to interpret qualitative data, did the authors use reflexivity to scrutinize their own biases when analyzing the qualitative data?
  2.     “The creation of the interview script by all the authors of the article” As previously mentioned, it is essential for the authors to address researcher positionality, which includes the background and experiences of the researchers relevant to the study topic.
  3.     “The validation and piloting of the interviews” How did the authors validate the interview scripts or the interviews? Piloting an interview script does not necessarily mean validation. Please exercise caution when using the term "validation."
  4.     “The rigorousness of the analysis techniques and the systematic process of transcription.” 

After reading the data analysis section, I found it difficult to comprehend the specific steps used by the authors to analyze the data and establish the code system. Did the authors use the code system to code the data? Or? Please explain. 

  1.     “Reading of all the results and analysis, and the detailed information about the methodological process followed to facilitate the transferability of the study” I do not understand what it means. 

 As this last idea is confusing, we have removed it from the manuscript.

Discussion and Conclusions:

14- The discussion is lengthy. The authors kept highlighting the point regarding the fact that some students had to change their educational modality during secondary school. However, this particular point seems to be unrelated to the main focus of the study (I believed the main focus of the study was centered around post-secondary options). To improve the clarity and relevance of the study, I would recommend that the authors provide a concise description that highlights the implications for secondary school, while shifting the primary focus towards university and job placement programs. I found the discussion on external factors and family support to be engaging and enjoyable to read.

Thank you for your suggestions. Based on them, we have focused the discussion on the key aspects that are related to the objectives of the research.

Reviewer 4 Report (New Reviewer)

The article addresses an interesting subject in the field of intellectual disabilities, since it is not only of interest but also a relevant topic in terms of the realization of the rights of people with DID.

It also raises an important issue that is related to the lack of coordination between secondary and university education, an issue that must be considered when facing the new challenges of educating people with DID.

Regarding the type of study, the approach, the questions that guide the investigation and the procedure are clear, if it is not clear how the issue of informed consent was worked on, or if the study went through a bioethics committee.

As for the results, answers are given to all the questions that were raised, however, it would have been interesting to advance in these insofar as not only presenting descriptions of what people raise, but also being able to go a step further and be able to relate the themes addressed in order to delve into each of the questions raised in the investigation.

As for the conclusions, although they address relevant issues, it is necessary to complement them with elements referring to the rights of people with disabilities, how a university education is or is not a guarantee to be able to face the world of work.

It would also be interesting to relate the results on how the educational trajectory of people impacts, with external factors, or how job profiles are addressed with the reasons for studying at the university.

Studies like the present one are a contribution to the development of knowledge in the field of disability, it is also salvageable since it listens to its protagonists, however it would be good to enrich how the results are presented

Author Response

Response to Reviewer

Thank you very much. We appreciate your comments on how to improve the paper. Your suggestions will be answered below.

Point 1: Regarding the type of study, the approach, the questions that guide the investigation and the procedure are clear, if it is not clear how the issue of informed consent was worked on, or if the study went through a bioethics committee.

Response 1: Based on your suggestions, the "Ethical aspects of research" section has been improved. The certificate from the ethics committee has also been sent to the editor of the journal.

Point 2: As for the results, answers are given to all the questions that were raised, however, it would have been interesting to advance in these insofar as not only presenting descriptions of what people raise, but also being able to go a step further and be able to relate the themes addressed in order to delve into each of the questions raised in the investigation.

Response 2:  Your feedback is greatly appreciated. We believe that the Discussion already provides a comprehensive explanation of these terms and that their inclusion in this section may be redundant. We have previously been advised to maintain a results-only approach in the Results section.

Point 3: As for the conclusions, although they address relevant issues, it is necessary to complement them with elements referring to the rights of people with disabilities, how a university education is or is not a guarantee to be able to face the world of work. It would also be interesting to relate the results on how the educational trajectory of people impacts, with external factors, or how job profiles are addressed with the reasons for studying at the university.

Response 3: Thank you for your suggestions. They have been included in the conclusions.

Round 2

Reviewer 1 Report (Previous Reviewer 1)

The authors made some theoretical improvements, but the research methodology is still suffering. The opinion for research improvement was clear and not so difficult to implement it. But it would be something interesting for readers, and for the persons interested in this topic. The authors did not made an improvement for the research, by using a quantitative method, thus, the article is rejected from publication

Moderate editing of English language required

Author Response

Thank you for your comments. We will keep them in mind for future research.  
Best regards.

Reviewer 3 Report (New Reviewer)

The authors have been responsive to the initial review of the manuscript and have added additional citations and clarified information about the context of the study. These responses to the original critiques have strengthened the manuscript. 

I suggested the authors should seek English editing services. 

Author Response

Thank you for your suggestions. They helped us improve the article. As for English, the document was translated by an official translator and then proofread by another translator, following your recommendation. We have sent the translation certificate to the publisher. 

This manuscript is a resubmission of an earlier submission. The following is a list of the peer review reports and author responses from that submission.

Round 1

Reviewer 1 Report

The study has an interesting theme, but the methodology is flawed, with reference to the following points. In the Introduction, at the end, the brief introduction of each chapter, the limitations of the research and the novelty are missing. The methodology is too simple. It should include, along with the research questions, the aim, objectives and research hypothesis. The results only cover part of the answers, based on a qualitative study. In order to be published, the study must also contain a quantitative study. The authors must develop a questionnaire and use R-squared between the variables analysed, the regression function to measure the impact of each factor presented as having influence. Also, ANOVA or another method. The conclusion and discussion should have implications for students, universities, teachers, parents and society. Also missing are boundaries and future directions. In this form, I propose that this article be rejected.

Extensive editing of English language required

Reviewer 2 Report

Dear author(s)

It was my pleasure to review your manuscript entitled “Motives and external factors that influence access to university and job placement programs for individuals with intellectual disability” and advise you to prosper your current research project. In my view, your topic has touched on a critical issue in a fascinating context. However, there are many spaces to be improved in terms of argumentation, theoretical background, research method, and findings. I hope my below comments would help you develop your work into groundbreaking research in your domain.

The positioning of the paper is not entirely clear. It is better to explain the gap in this article further.

The introduction should clearly illustrate (1) what we know (the key theoretical perspectives and empirical findings) and what we do not know (major, unaddressed puzzle, controversy, or paradox does the study address, or why it needs to be addressed and why this matters) and (2) what we will learn from the study, and how the study fundamentally changes, challenges, or advances scholars’ understanding. Much sharper problematization is required so that the introduction draws the reader into the paper. At the end of the introduction, we should have a clear idea of what the paper is about (i.e., its motivation, the gap in understanding that the paper is trying to address, and a summary of theoretical contributions).

Paragraph 3 explains what we need to find out.

Paragraph 4 explains briefly what this paper will do to find out, the method, etc.

Paragraph 5, with no references, explains the structure of this paper.

This is one of the most critical parts of the paper that found lacking detail.

The method should be adequately described to show how the research was conducted to improve clarity and transparency.

What were the reasons for using the method?

How is validity and reliability done?

In the discussion section, for each case study that you have identified separately, the results should be written, what effects it has on the main result? 

What are the results of your research, and how can it help your community?

It is better to add these two parts (Limitation and future directions) after the conclusion.

Using the following reference could be beneficial as these add more evidence to the literature review section:

(2022). Investigating social capital, trust and commitment in family business: Case of media firms. Journal of Family Business Management, 12(4), 938-958.

Best of luck with the further development of the paper.

It needs minor revision.

Reviewer 3 Report

The authors clearly present the normative and human rights frameworks for the educational inclusion of students with disabilities in higher education. Although no statistical values are presented on the number of students with disabilities in higher education (which would significantly place the social problem of the phenomenon that the study addresses), it is clear that this is an "invisible" population for which there are no well-elaborated educational policies aimed at this type of population. The methodological proposal is appropriate and the main questions clearly represent the topic addressed in the work; however, throughout the presentation of the results, many statements are extracted that are not always evidenced by the excerpts from the interviews of the participants.

A topic that is not addressed in the introduction of the paper, nor in the discussion is the continuous teacher training aimed at inclusive education, so that the conclusions reached make suggestions of the type of educational policies at an institutional level but not within the classroom as it would correspond to what is expected of a teacher in front of a group when people with disabilities are included in regular classrooms.